

# Impact of online support of physical activity management using a wearable device on renal function in patients with acute coronary syndrome: a randomized controlled trial protocol

Toshimi Sato[1], Daisuke Suzuki[2], Yuichiro Sasamoto[3], Masahiro Ono[4], Namiko Shishito[4], Kohko Kanazawa[5], Akihito Watanabe[3], Koichi Naito[6], Shinichiro Morishita[1] and Masahiro Kohzuki[7,8]

[1] Department of Physical Therapy, School of Health Sciences, Fukushima Medical University, Fukushima, Japan
[2] Department of Rehabilitation, Southern Tohoku General Hospital, Koriyama, Japan
[3] Department of Rehabilitation, Ohta Nishinouchi Hospital, Koriyama, Japan
[4] Department of Cardiology, Southern Tohoku General Hospital, Koriyama, Japan
[5] Department of Cardiology, Ohta Nishinouchi Hospital, Koriyama, Japan
[6] Faculty of Medical Science, Nagoya Women's University, Nagoya, Japan
[7] Yamagata Prefectural University of Health Sciences, Yamagata, Japan
[8] Tohoku University Graduate School of Medicine, Sendai, Japan

Corresponding author
Toshimi Sato, satot-pt@fmu.ac.jp

## ABSTRACT

**Background.** Acute coronary syndromes (ACS) often cause rapid decline in renal and cardiac function. In patients with ACS, combined renal dysfunction is associated with increased overall mortality and cardiovascular events. Physical activity (PA) management may crucially contribute towards protection of renal function in patients with ACS. This article describes the study protocol of a randomized controlled trial (RCT) assessing whether online support for PA management using wearable devices and information communication technology for patients with ACS facing difficulties in participating in outpatient cardiac rehabilitation after discharge can protect renal function following disease onset.

**Methods.** We have designed a two-arm RCT with a 3-month follow-up period. The online support intervention incorporates monitoring of PA, pulse rate, and blood pressure using a wearable device with an accelerometer and a web application system, as well as periodic educational feedback and goal setting. The primary study endpoint is the estimated glomerular filtration rate based on serum cystatin C (eGFRcys). The intervention effect will be assessed using the eGFRcys at 3 months adjusted for baseline values. The secondary endpoints are the urine albumin/creatinine ratio, brain natriuretic peptide levels, average step count, peak oxygen uptake, quality of life, and incidence of adverse events.

**Discussion.** This RCT will provide evidence regarding the effectiveness of online support for PA management as a renal protection strategy following ACS onset. This novel strategy not only mitigates barriers impeding participation in outpatient cardiac rehabilitation and protects cardiac and renal function in patients with ACS, but also may contribute towards improving survival and recurrence rates, preventing dialysis, and reducing medical and long-term care costs.

**Trial registration:**. The trial was registered in the Japan Registry of Clinical Trials on July 5, 2024. The registration number is jRCT1022240014 (Impact of Online Support of Physical Activity Management Using a Wearable Device on Renal Function in Patients with Acute Coronary Syndrome).

# INTRODUCTION

Acute coronary syndromes (ACS), including acute myocardial infarction, often cause rapid decline in renal and cardiac function (*Hillege et al., 2003*; *Eijkelkamp et al., 2007*; *Esmeijer et al., 2018*). Further, approximately 30% of patients with ACS develop new-onset renal dysfunction, with 40% of them progressing to chronic kidney disease (CKD) (*Marenzi, Cosentino & Bartorelli, 2015*). Among patients with ACS, combined renal dysfunction is associated with increased overall mortality and cardiovascular events (*Anavekar et al., 2004*; *Okina et al., 2021*). Therefore, it is important to maintain both cardiac and renal function after onset of ACS. However, there remain no established interventions for preventing renal dysfunction after onset of ACS.

Cardiac rehabilitation (CR) has been shown to improve exercise capacity, coronary risk factors, and quality of life (QOL), as well as reduce cardiovascular recurrence and overall mortality. Further, numerous guidelines recommend CR as a Class I procedure for managing ST-elevation myocardial infarction (*O'Gara et al., 2013*; *Amsterdam et al., 2014*; *Ibanez et al., 2018*; *Kimura et al., 2019*; *Makita et al., 2022*). Moreover, recent studies have demonstrated that CR centered on supervised exercise program maintained or improved renal function in patients with ACS (*Takaya et al., 2014*) or cardiovascular disease (CVD) (*Toyama et al., 2010*; *Fujimi et al., 2015*; *Iso et al., 2015*; *Kimura et al., 2015*; *Hama et al., 2018*; *Sasamoto et al., 2021*; *Hama et al., 2022*). *Takaya et al. (2014)* reported that late phase II CR of patients with ACS and comorbid CKD improved exercise tolerance, cardiac-related markers such as B-type natriuretic peptide (BNP), and estimated glomerular filtration rate (eGFR). Similarly, several observational studies have demonstrated that CR helps maintain or improve eGFR or urinary protein levels in patients with CVD (*Toyama et al., 2010*; *Fujimi et al., 2015*; *Iso et al., 2015*; *Kimura et al., 2015*; *Hama et al., 2018*; *Sasamoto et al., 2021*; *Hama et al., 2022*). This suggests that renal function in patients with CVD may be a novel therapeutic target for CR; however, this remains to be validated by randomized controlled trials.

Previously, we have examined the association between physical activity (PA) and changes in renal function in patients with ACS undergoing late phase II CR and found that a daily step count (approx. 5,000 steps/day) and moderate-to-vigorous PA (>3.0 METs) could improve the eGFR (*Sato et al., 2019*; *Sato et al., 2021*; *Sato et al., 2024*). Moreover, PA has been shown to improve renal function in patients with ACS with and without complicated CKD. Furthermore, exercise intolerance in patients with ACS is an independent predictor

of more rapid decline in renal function (*Sato et al., 2023*). Accordingly, CR interventions that improve exercise tolerance and PA may be effective strategies for protecting renal function.

However, there are serious limitations in providing outpatient CR after the onset of ACS. For example, the participation rate in phase II CR among outpatients after ACS in Japan is extremely low at 18% (*Arakawa et al., 2016*). Factors that influence participation in exercise programs include external factors (safety, transportation, social support), internal factors (physical function, cognitive function, and emotions), and cultural factors (*Campkin, Boyd & Campbell, 2017*). In Japan, the percentage of patients with ACS who were recommended CR by their primary physician is as low as 32% (*Ohtera et al., 2017*). Despite the high level of evidence for CR, the low participation rate of CR has remained unchanged worldwide.

On the other hand, PA monitoring interventions based on wearable devices such as pedometers have been found to contribute to improvements in PA and mitigating cardiovascular risk in patients with ACS (*Houle et al., 2011*; *Houle et al., 2013*). Recent reports have described the effectiveness of remote CR using wearable devices as an alternative to traditional center-based CR. Remote CR, which allows participation in CR without hospital visits, has several advantages such as eliminating the time required to visit the hospital, solving access problems among patients living far from the hospital, potentially reducing costs, avoiding the risk of infection during group exercise, and conferring the benefits of information technology equipment (*Nakayama et al., 2023*). Previous meta-analyses have shown that remote CR can effectively improve exercise tolerance and prognosis, and it is equally or more effective in improving PA levels, QOL, adherence, lipid metabolism, blood pressure, and hospital readmissions compared with center-based CR (*Rawstorn et al., 2016*; *Ramachandran et al., 2022*; *Zhong et al., 2023*). Specifically, remote PA monitoring using wearable devices may be a novel CR strategy that can be adopted by more patients with ACS since it eliminates time and location constraints, as well as provides real-time patient information and objective feedback. This intervention has the potential for effectively improving the motivation and adherence in patients with ACS to PA management. However, the effect of remote support for PA management using wearable devices on improving renal function in patients with heart disease remains unclear.

We hypothesize that remote support for PA management using wearable devices and information communication technology for patients with ACS who have difficulty participating in outpatient CR may protect renal function after the onset of ACS.

## Objective

The primary objective of this study is to determine the impact of remote support for PA management using a wearable device on renal function in patients with ACS. The secondary objectives are to determine the impact of this intervention on other factors such as cardiac and physical function as well as QOL, and to determine whether it can be an alternative intervention for patients with ACS who have difficulty participating in outpatient CR.

### Clinical implications

This study will provide evidence regarding the effectiveness of remote support for PA management in renal protection after ACS, which may improve survival and recurrence rates, prevent dialysis, and reduce medical and nursing care costs for patients with ACS. The validation results of this study will provide important scientific information for establishing a novel CR strategy that can be adopted by many patients with ACS who have difficulty participating in outpatient CR.

## METHODS

### Study design

The study design will be an open-label, two-arm, parallel, randomized control trial (Fig. 1). The study protocol conforms to the 2017 CONSORT NPT Extension and the study intervention is described in accordance with the CONSORT-EHEALTH.

### Ethics approval

This study received approval from the Ethics Committee of Fukushima Medical University (Approval No. REC2023-174).

### Study eligibility and recruitment

The inclusion criteria for patients in this study are as follows: (i) patients with ACS who underwent percutaneous coronary angioplasty and inpatient CR, (ii) age ≥18 years at the time consent was obtained, (iii) inability to participate in outpatient CR after discharge, (iv) having a smartphone, and (v) agreeing to participate in this study after receiving verbal and written explanations from the researchers. Written consent was obtained from all participating patients in this study. The exclusion criteria are as follows: (i) lack of independence in activities of daily living, (ii) unstable angina, (iii) uncontrolled arrhythmia causing hemodynamic abnormalities, (iv) severe valvular disease, (v) uncontrolled diabetes, (vi) other diseases that contraindicate the use of exercise therapy, (vii) undergoing dialysis, (viii) other acute diseases or indications for surgical treatment, (ix) dementia, and (x) inability to make regular outpatient visits to the study site after discharge.

Researchers in Southern Tohoku General Hospital and Ohta Nishinouchi Hospital, which are the institutions conducting the study, will recruit study participants who have the potential to meet the eligibility criteria. The principal investigator of Fukushima Medical University, which is the institution responsible for the research, will oversee and manage patient enrollment.

### Randomization and blinding

Randomization allocation will be performed on the web using the UMIN INDICE cloud (https://www.umin.ac.jp/indice/cloud.html). Allocation will be completed within 2 weeks of initiation of inpatient CR. This study will not be blinded. Adjustment factors for allocation will be age (<60 years *vs.* ≥60 years), sex (male *vs.* female), and highest creatine kinase level (<3,000 U/L *vs.* >3,000 U/L).

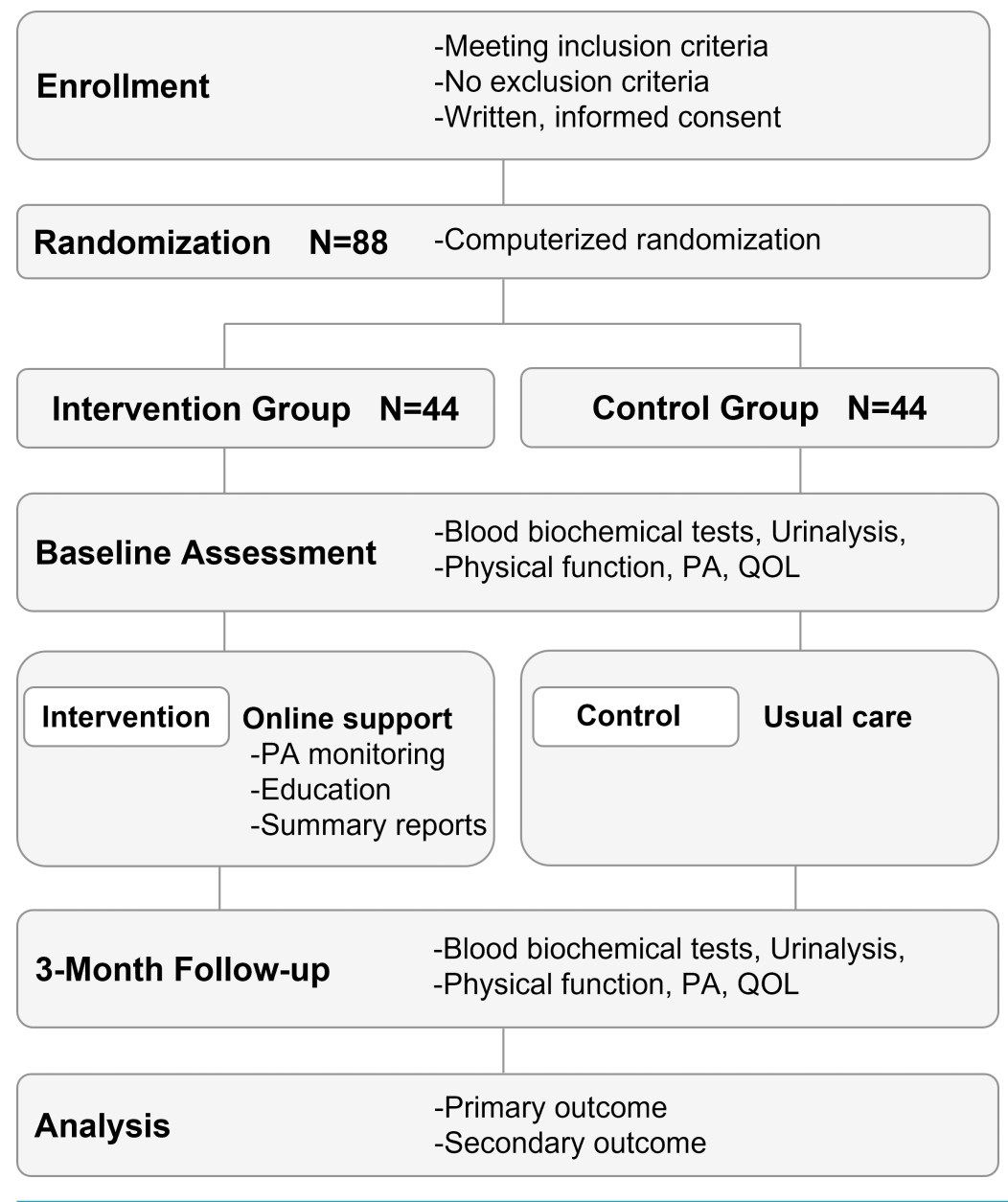

**Figure 1** **Randomized controlled trial design and flowchart.** Patients with ACS who cannot participate in center-based CR for any reason are enrolled. Patients in both groups receive usual care. The online support intervention incorporates monitoring of physical activity, pulse rate, and blood pressure using a wearable device with accelerometer and a web application system, as well as periodic educational feedback and goal setting. ACS, acute coronary syndromes; CR, cardiac rehabilitation.

## Intervention

Online support for PA management will begin at the time of discharge; additionally, patients in the intervention group will manage their own PA based on educational support provided by the physical therapist under the supervision of the primary physician (Fig. 2). The physical therapist administering the intervention must have ≥1 year of experience in

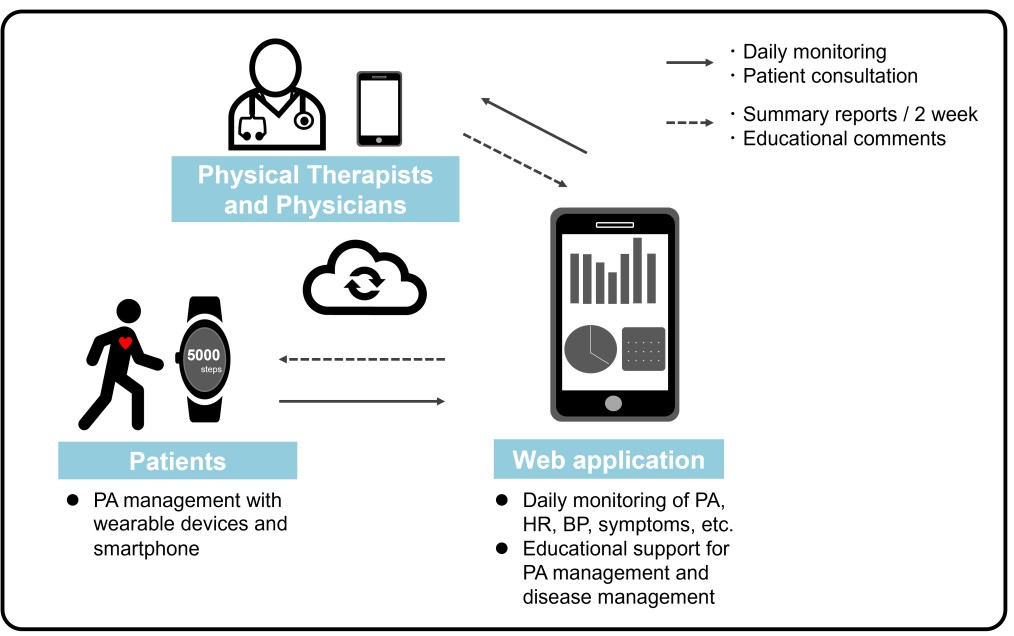

**Figure 2 Online support of physical activity management.** Online support of PA management in this study will begin at the time of discharge from the hospital, and patients in the intervention group will manage their own PA based on educational support by the physical therapist under the supervision of the primary physician. PA, physical activity.

physical therapy for patients with CVD, including ACS, or be a registered instructor of CR by the Japanese Association of Cardiac Rehabilitation (https://www.jacr.jp/en/).

## Online support for PA management

Patients in the intervention group will receive a wrist-worn wearable device (Fitbit Inspire 3, Fitbit, Inc, San Francisco, CA, USA) after assignment, and a web application (Recaval, SapplyM, Inc., Tokyo, Japan) for PA management will be downloaded to their smartphones. Data regarding PA and pulse rate obtained from the wearable device will be transmitted in real time *via* the application to the tablet device of the physical therapist in charge of the research facility. The wearable device is designed to locally store data until an internet connection is available. In addition to this biometric monitoring, the web application has chat and report feedback functions that allow two-way communication between patients and medical professionals. Patients will be instructed to wear the wearable device unless they are bathing or charging the device. Moreover, patients will be instructed to enter their daily blood pressure and self-records of subjective symptoms into the application, which will also be monitored daily by the physical therapist. Using this web application, patient information is completely anonymized; each patient is assigned a specific ID. In addition, this web application has a robust access restriction function. Only the interventionists at each research facility are only given access to the patient data of their own institution and have no access to the patient data of other research institutions.

**Table 1  Criteria for discontinuation of the intervention.**

| Criteria |
| --- |
| (1) Cardiac arrest, severe bradycardia, fatal arrhythmia (ventricular tachycardia/ventricular fibrillation), or symptoms that cannot be ruled out during follow-up |
| (2) In case of sudden deterioration of vital signs or subjective symptoms (severe chest pain, abdominal pain, back pain, epileptic seizure, loss of consciousness, hypotension, severe joint pain, muscle pain, *etc.*). |
| (3) Accidents (falls, falls, bruises, trauma, *etc.*) |
| (4) In case of persistent hypertension (systolic blood pressure $\geq$ 250 mmHg or diastolic blood pressure $\geq$ 115 mmHg) or hypotension (systolic blood pressure < 80 mmHg) that is difficult to control. |
| (5) Deterioration of consciousness during follow-up |
| (6) The attending physician determines that the intervention should be discontinued due to changes in the patient's condition or treatment strategy during follow-up |
| (7) The patient is hospitalized |
| (8) The patient wishes to discontinue the intervention. |

At the beginning of the intervention and at 2-week intervals thereafter, the physical therapist will provide the patient with a summary report of daily vital sign data, including educational comments and praise commending the achievement of PA and activity pulse rate goals, through the app. Regarding the target PA value, which is the basis of the educational support, the patients will be assigned a target of $\geq$5,000 daily steps, referring to a previously reported cutoff value for the number of steps per day that affects improvement of renal function in patients with ACS (*Sato & Kohzuki, 2021*; *Sato et al., 2021*). Further, this target can be achieved without worsening of vital signs or subjective symptoms. For example, if the weekly average daily step count is <5,000 steps/day, a smaller target of 1,000 steps/day is set for the next week and thereafter, which allows the patient to achieve the goal of $\geq$5,000 steps/day in incremental stages. As a safety management measure, the target values for pulse rate during activity and duration of activity above moderate intensity are set according to each patient's results of the baseline cardiopulmonary exercise test and Karvonen's formula (*Makita et al., 2022*). Table 1 shows the criteria for discontinuation of the intervention.

Additionally, while monitoring daily PA, pulse rate, and self-recorded data regarding blood pressure, body weight, and subjective symptoms, the physical therapist will check the status and provide educational support for PA modification using the chat function of the application as required in case of the following scenarios:

- The pulse rate during activity exceeds the target pulse rate by 20 beats/minute for >10 min
- The pulse rate during PA falls below 50% of the target value for >5 consecutive days
- The patient does not wear an activity monitor for >5 consecutive days
- The patient's self-recorded data indicate worsening subjective symptoms, including chest pain, shortness of breath, or fatigue
- The patient's has questions or complaints regarding PA management or disease management
- The physiotherapist or physician in charge considered it necessary.

Furthermore, interviews will be conducted by telephone or *via* the ZOOM web conferencing service, as necessary.

## Usual care

All patients in both groups will receive usual care, including patient education, medications, outpatient visits, and laboratory tests upon discharge, within the relevant scope. However, if patients wish to participate in outpatient CR during the observation period, their wishes will be respected, and they will be allowed to participate in outpatient CR after consultation with their physician. In this case, the patient will not be excluded from the study and will continue to receive "intervention and outpatient CR" in the intervention group and "usual care and outpatient CR" in the control group.

## Outcome assessment

Table 2 describes the outcome measures for this trial. Patient characteristics and baseline measurements will be assessed at discharge (~2 weeks after the onset of ACS) and reassessed after 3 months.

The primary outcome of this study will be the eGFR based on serum cystatin C (eGFRcys). In general clinical practice, the eGFR based on serum creatinine is widely used as an index for assessing renal function. However, PA, including exercise, can alter serum creatinine levels *via* changes in skeletal muscle mass. Therefore, cystatin C, which is independent of skeletal muscle mass, is preferred when examining the effects of PA on renal function (*Séronie-Vivien et al., 2008*; *Poortmans et al., 2013*). The eGFRcys will be estimated using the Japanese Society of Nephrology equation as follows:

eGFRcys (mL/min/1.73 m$^2$) = 104 × Cystatin C$^{-1.019}$ × 0.996$^{Age}$ × 0.929 (if female)–8 (*Horio et al., 2013*).

This study will perform between-group comparisons of eGFRcys at 3 months adjusted for baseline values. eGFR decline after ACS may be particularly steep during the first 90 days after onset (*Hillege et al., 2003*). It is important to examine the inhibitory effect on eGFR decline during this period; accordingly, we set a 3-month follow-up period for this study. Most previous studies on the impact of outpatient CR on eGFR have similarly set a 3-month intervention and observation period (*Toyama et al., 2010*; *Takaya et al., 2014*; *Hama et al., 2018*; *Sasamoto et al., 2021*; *Hama et al., 2022*). Therefore, the follow-up period can be considered reasonable.

Secondary outcomes will include the urine albumin/creatinine ratio (ACR), BNP, average step count, peak oxygen uptake (VO2) based on a symptom-limited cardiopulmonary exercise test (CPET), Euro QOL 5-dimensions 5-levels, and incidence of adverse events. In addition, blood biochemical analysis data such as urea nitrogen; albumin; triglycerides; high- and low-density cholesterol; hemoglobin; and blood glucose will be collected. These outcome measures will be assessed concurrently with the primary evaluation. In the symptom-limited CPET, to avoid overloading the cardiovascular system after ACS, the exercise load is terminated when the gas exchange ratio reaches 1.2, which is defined as the peak point, even before reaching the symptom limit (*Sato et al., 2023*).

PA assessment will be performed immediately after discharge and after 3 months for a 7-day period. The Active Style Pro HJA-750 (Omron Healthcare Company; Kyoto,

**Table 2  All outcome measures and assessment schedule for this study.** CPET, cardiopulmonary exercise test; EQ-5D-5L, Euro QOL 5-dimensions 5-levels; eGFRcreat, estimated glomerular filtration rate calculated from serum creatinine; eGFRcys, estimated glomerular filtration rate calculated from serum cystatin C; LPA, light-intensity physical activity; MPA, moderate-intensity physical activity; QOL, quality of life; SB, sedentary behavior; VPA, vigorous-intensity physical activity.

| | Baseline | 3-month |
|---|:---:|:---:|
| **Characteristics**: sex, age, alcohol consumption history, smoking history, location of coronary artery stenosis, Killip classification, contrast fee, medical history, comorbidities, echocardiography, length of hospital stay, height | ● | |
| **Body conditions**: weight, body mass index, resting blood pressure, resting pulse rate | ● | ● |
| **Physical function**: CPET and grip strength | ● | ● |
| **Blood biochemical tests**: BNP, urea nitrogen, cystatin C, creatinine, glomerular filtration rate (eGFRcys and eGFRcreat), albumin, triglycerides, HDL-cholesterol, LDL-cholesterol, hemoglobin, blood sugar | ● | ● |
| **Urinalysis**: Urine albumin/creatinine ratio | ● | ● |
| **Physical activity**: step count, SB, LPA, MPA, VPA | ● | ● |
| **QOL**: EQ-5D-5L | ● | ● |
| **Medication** | ←——————→ | |
| **Adverse events** | ←——————→ | |
| (Intervention group only) **Fitbit data**: daily sleep time and heart rate (maximum, minimum, and average) | ←——————→ | |

Japan), which has a 3-axis accelerometer that has been validated and shown to be highly reliable (*Oshima et al., 2010*; *Ohkawara et al., 2011*), will be used to evaluate PA and the average number of steps. Sedentary behavior, light-intensity PA, moderate-intensity PA, and vigorous-intensity PA will be defined as PA of ≤1.5 metabolic equivalents (METs), >1.5 and <3.0 METs, ≥3.0 and <6.0 METs, and >6.0 METs, respectively (*Bull et al., 2020*); moreover, the time spent wearing the device will be collected. To avoid the impact of checking PA on increasing PA levels, the researcher sets the device to prevent displaying the PA data. The patients will be instructed to wear the accelerometer at all times except when sleeping and bathing. If the time the device is worn is <10 h per day, activity data for that day will be excluded. The minimum number of days required for evaluation is defined as "4 days or more", which is more than half of the 7 days (Fig. 3). Patients will return the accelerometer to the hospital by mail at the end of that 7-day PA measurement period. The principal investigator at each hospital will transfer PA data from the accelerometer to the computer *via* a reader (HHX-IT4, Omron Healthcare Company; Kyoto, Japan).

## Other parameters

To explain the baseline characteristics of the patients, we will investigate information on sex, age, drinking history, smoking history, site of coronary artery stenosis, Killip classification, contrast agent dose, medical history, complications, length of hospital stay, height, and weight from the electronic medical record at the time of registration. Moreover, we will

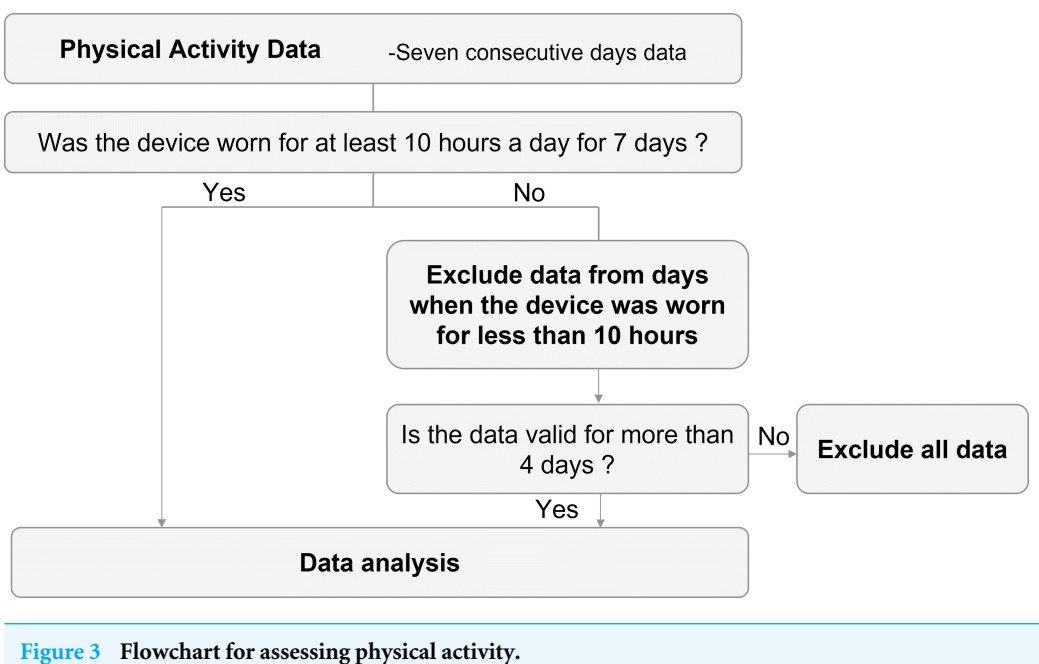

**Figure 3** Flowchart for assessing physical activity.

investigate the left ventricular ejection fraction evaluated by echocardiography at the time of discharge.

## Statistical analysis

In this study, the analysis will involve an intention to treat analysis. Therefore, patients' consent will be obtained to use 3-month post-assessment information derived from the observed clinical data in the event of dropout or poor adherence in the intervention arm. Regarding background information of participants, discrete data, including frequencies, proportions, and 95% confidence intervals, will be calculated for each group. For continuous data, the mean, standard deviation, and 95% confidence interval will be calculated for each group.

The primary outcome, eGFRcys, will be examined using analysis of covariance (ANCOVA), with adjustment for baseline values as covariates for the least squares mean of eGFRcys at 3 months. Secondarily, between-group comparisons of the change in eGFRcys will be performed in the same way.

The secondary endpoints of ACR, BNP, average number of steps, peak VO2, and EQ-5D will be analyzed using ANCOVA, with adjustment for baseline values. Between-group comparisons of the incidence of adverse events will be performed using the $\chi$-square test.

All statistical analyses will be performed using SPSS version 29.0 (IBM Corp., Armonk, NY, USA). Statistical significance will be set at an alpha-level of 0.05.

## Sample size calculation

To determine the impact of online PA management support on eGFRcys, we will examine between-group differences in eGFRcys. Since there have been no reports of a clinically meaningful minimum difference in eGFRcys among patients with ACS, we

arbitrarily assumed that the between-group difference in eGFRcys after 3 months would be five mL/min/1.73 m$^2$. If the standard deviation of eGFRcys in both groups is set to be 16.0 in reference to our previous study (*Sato et al., 2019*), the effect size was 0.3125.

Subsequently, assuming that between-group comparisons of eGFRcys after 3 months would be performed using ANCOVA with baseline values as covariates, G*Power 3.1 was used to calculate the required sample size with an effect size $f = 0.3125$, alpha $= 0.05$, and power $= 0.8$. The results showed that 83 cases were detected. Accordingly, we calculated that the required sample size was 83 cases, and the target number of cases was set at 88 (44 in the intervention group and 44 in the control group), taking into account the possibility of dropouts (5%) in both groups.

## DISCUSSION

This randomized controlled trial will provide evidence regarding the effectiveness of online support for PA management in renal protection after ACS. In patients with ACS, combined renal dysfunction has been associated with increased overall mortality and cardiovascular events.

Previous studies have shown that PA management (*Sato et al., 2019*; *Sato & Kohzuki, 2021*; *Sato et al., 2021*; *Sato et al., 2024*) and outpatient CR (*Takaya et al., 2014*) may reduce renal function decline in patients after ACS. However, this evidence remains uncertain since it has been exclusively obtained from observational studies. Furthermore, the low CR participation rate of patients with ACS in practice has impeded the feasibility of PA management as a renal protection strategy.

The ultimate goal of this study is to develop a more feasible renal protection strategy that addresses these challenges. Therefore, we focused on wearable device and web application technologies to design an online support intervention of PA management suitable for renal protection in patients with ACS. The strength of this intervention is its ability to monitor daily PA, pulse, blood pressure, and other biometric information in real time as well as to provide educational feedback and goal setting based on this information. This allows patients with ACS to engage in PA improvement with remote support from a physical therapist or physician without having to visit the hospital, and may be an alternative to the late phase II CR. This novel strategy not only removes barriers impeding outpatient CR participation and protects cardiac and renal function in patients with ACS, but also may improve survival and recurrence rates, avoid dialysis, and reduce medical and long-term care costs.

In addition, the primary endpoint in this study is the eGFRcys, which is less sensitive to skeletal muscle mass and diet. This will minimize the possibility that changes in skeletal muscle mass will mask the relationship between increased PA and changes in renal function, and therefore enhance the scientific nature of our findings.

## ACKNOWLEDGEMENTS

The authors would like to thank all patients whose participation make this investigation possible and all colleagues in Southern Tohoku General Hospital and Ohta Nishinouchi Hospital for their contribution to the medical care of the patients.

### Funding

This work was supported by the Japan Society for the Promotion of Science Grants-in-Aid for Scientific Research (JSPS KAKENHI Grant) numbers JP 23K19837 received by Dr. Toshimi Sato, and JP24K14350 received by Dr. Masahiro Kohzuki. The funders had no role in study design, data collection and analysis, decision to publish, or preparation of the manuscript.

### Grant Disclosures

The following grant information was disclosed by the authors:
Japan Society for the Promotion of Science Grants-in-Aid for Scientific Research (JSPS KAKENHI Grant): JP 23K19837, JP24K14350.

### Competing Interests

The authors declare there are no competing interests.

### Author Contributions

- Toshimi Sato conceived and designed the experiments, performed the experiments, analyzed the data, prepared figures and/or tables, authored or reviewed drafts of the article, and approved the final draft.
- Daisuke Suzuki conceived and designed the experiments, performed the experiments, authored or reviewed drafts of the article, and approved the final draft.
- Yuichiro Sasamoto conceived and designed the experiments, performed the experiments, authored or reviewed drafts of the article, and approved the final draft.
- Masahiro Ono performed the experiments, authored or reviewed drafts of the article, and approved the final draft.
- Namiko Shishito performed the experiments, authored or reviewed drafts of the article, and approved the final draft.
- Kohko Kanazawa performed the experiments, authored or reviewed drafts of the article, and approved the final draft.
- Akihito Watanabe performed the experiments, authored or reviewed drafts of the article, and approved the final draft.
- Koichi Naito analyzed the data, authored or reviewed drafts of the article, and approved the final draft.
- Shinichiro Morishita analyzed the data, authored or reviewed drafts of the article, and approved the final draft.
- Masahiro Kohzuki analyzed the data, authored or reviewed drafts of the article, and approved the final draft.

### Clinical Trial Ethics

The following information was supplied relating to ethical approvals (i.e., approving body and any reference numbers):

This study received approval from the Ethics Committee of Fukushima Medical University (Approval No. REC2023-174).

### Data Availability

This paper is a research protocol paper.

### Clinical Trial Registration

The following information was supplied regarding Clinical Trial registration:

jRCT1022240014 and UMIN000053775

### Supplemental Information

Supplemental information for this article can be found online at http://dx.doi.org/10.7717/peerj.19067#supplemental-information.

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
