# Peer review of "Impact of online support of physical activity management using a wearable device on renal function in patients with acute coronary syndrome: a randomized controlled trial protocol"

_PeerJ, doi:10.7717/peerj.19067_

## Round 0.1 · original submission · Major Revisions

Dear Dr. Sato,

Your manuscript entitled " Impact of online support of physical activity management using a wearable device on renal function in patients with acute coronary syndrome: a randomized controlled trial protocol", which you submitted to PeerJ, has been reviewed by the editor and 3 experts in the field.

The reviewers are generally favorable but have raised significant concerns that must be addressed before the manuscript can be considered further. I would be willing to reconsider if you wish to undertake major revisions and resubmit.

If you decide to resubmit the revised version, please summarize all the improvements made in the new version and give answers to all critical points raised in the reviewers’ report in an accompanying letter. Copy and paste each and every reviewer's comment above your response. Please consider these points carefully, as the revised manuscript will undergo a second round of review by the same reviewers.

I hope you will be prepared to make the necessary amendments and submit a revised manuscript with a statement of how you responded to the reviewers’ comments.

Yours sincerely,

Stefano Menini

·

Basic reporting

The language used is clear and professional.

Good cover of the literature. Authors might want to include these early studies about ACS PA home monitoring.

1. Houle J, Doyon O, Vadeboncoeur N, Turbide G, Diaz A, Poirier P. Innovative program to increase physical activity following an acute coronary syndrome: randomized controlled trial. Patient Educ Couns. 2011 Dec;85(3):e237-44. doi: 10.1016/j.pec.2011.03.018. Epub 2011 May 4. PMID: 21546203.
2. Houle J, Valera B, Gaudet-Savard T, Auclair A, Poirier P. Daily steps threshold to improve cardiovascular disease risk factors during the year after an acute coronary syndrome. J Cardiopulm Rehabil Prev. 2013 Nov-Dec;33(6):406-10. doi: 10.1097/HCR.0000000000000021. PMID: 24104407.

The flow chart (figure 1) could benefit from some tidiness, are there a follow up echocardiography? If not, why it is not being repeated?
Table 1 should have a header row.

Hypothesis is well stated and seems testable.

Experimental design

The research is within the aims and scope of the journal.

The knowledge gap is well identified and tries to answer a persisting question - the usefulness and effectiveness of remote monitoring.

Ethical approval provided is in Japanese language, perhaps, an English version would be important.

Methods:
Inclusion, exclusion and randomization are well described.

Regarding wearing compliance. How will you make sure that this a valid day? Would you discard any days where the watches were worn less than the expected? I recommend setting a certain criterion for example: minimum of 10 hours during active hours (thus exclude nighttime for example). Please remember that not all patients stick to the instructions and may wear it irregularly which may affect the quality of the data.

I also recommend a flow chart describing validity of any given wearing day.

How many days are required per week/month to count it valid?

Would you stop the watch notifications, to avoid extra activities?

How would you address the issue of not having consistent network? Please remember, there is a large number of the targeted population who are elderly, thus, the use of internet might not be always available to record their data in the web application.

You mentioned summary report will be sent to the patients. Would you address the implications of this report to the patient? This might induce alteration to their exercise capacity.

You did not address data confidentiality and anonymity, as you will be using a separate software developed by your research group.

There was no mention of using echocardiography in the baseline assessment, it was only shown in the figure. Would it be done at follow up? What are you measuring?

Validity of the findings

This is a study design paper, there was no results nor any statistical analysis. However, the authors have defined the novelty of this work.

Additional comments

No comment

·

Basic reporting

The article presents a good language style, without major errors, except in the use of punctuation as in line 198, where PA, pulse rate and self-recorded should be separated by as and not by a semicolon.

References are appropriate and within the scope of the article's topic.

The structure of the article is very good. However, the data, trend graph or captured data are not presented, to observe the range, if there are outlayers or something similar.

In general, the results are very descriptive and without showing relevant values ​​or data.

Experimental design

The research is original and is in the journal's scope. I think that this type of research may be of interest for the treatment of diseases such as ACS.

The objective of the article and its contribution to knowledge are well defined, according to the bibliographic review.

According to the authors' description, ethical aspects were taken into account and methodological work was carried out in accordance with what is established for this type of health work, taking into account informed consent and the health aspects of the patients.

The methods are clearly described, but reproducibility is not clear, since data, averages, standard deviations, etc., are not presented. Statistical methods are mentioned, but data and results are not presented.

Validity of the findings

The results are valid and could be replicated, although I consider the sample to be small. It would be good to know what the size or general population group is, and to know what percentage the sample represents of this population.

As mentioned before, the article does not show graphs of the data and trends.

Additional comments

Authors should present statistical results and how the sample size was selected.

·

Basic reporting

Line 59: The authors could use ~ instead of =.
Line 175: The use of double brackets seems to be a typo.
The other basic reporting guidelines are followed.

Experimental design

The experimental design is clear and reasonable.

Validity of the findings

This is a study protocol - therefore, this section does not apply.

Additional comments

The authors have described a clear protocol for a relatively small RCT assessing the impact of a wearable-based mobile health platform on renal function in patients with ACS. I wish them the best in conducting the trial.

---

## Round 0.2 · accepted · Accept

Dear Dr. Sato,

Thank you for submitting the revised version of your manuscript. I have personally reviewed the revision and verified that all the reviewers' comments have been adequately addressed. I am pleased to inform you that your manuscript is now ready for publication in PeerJ in its present form.

I want to thank all the reviewers for their efforts in improving the manuscript and the authors' cooperation throughout the review process.

Sincerely,
Stefano Menini

·

Basic reporting

All comments have been addressed

Experimental design

All comments have been addressed

Validity of the findings

All comments have been addressed